# EEG Emotion Recognition by Fusion of Multi-Scale Features

**DOI:** 10.3390/brainsci13091293

**Published:** 2023-09-07

**Authors:** Xiuli Du, Yifei Meng, Shaoming Qiu, Yana Lv, Qingli Liu

**Affiliations:** 1Communication and Network Laboratory, Dalian University, Dalian 116622, China; mengyifei0729@163.com (Y.M.); qiushaoming@dlu.edu.cn (S.Q.); lvyana0731@163.com (Y.L.); liuqingli@dlu.edu.cn (Q.L.); 2School of Information Engineering, Dalian University, Dalian 116622, China

**Keywords:** EEG, emotion recognition, multi-scale convolution, ECANet, deformable convolution

## Abstract

Electroencephalogram (EEG) signals exhibit low amplitude, complex background noise, randomness, and significant inter-individual differences, which pose challenges in extracting sufficient features and can lead to information loss during the mapping process from low-dimensional feature matrices to high-dimensional ones in emotion recognition algorithms. In this paper, we propose a Multi-scale Deformable Convolutional Interacting Attention Network based on Residual Network (MDCNAResnet) for EEG-based emotion recognition. Firstly, we extract differential entropy features from different channels of EEG signals and construct a three-dimensional feature matrix based on the relative positions of electrode channels. Secondly, we utilize deformable convolution (DCN) to extract high-level abstract features by replacing standard convolution with deformable convolution, enhancing the modeling capability of the convolutional neural network for irregular targets. Then, we develop the Bottom-Up Feature Pyramid Network (BU-FPN) to extract multi-scale data features, enabling complementary information from different levels in the neural network, while optimizing the feature extraction process using Efficient Channel Attention (ECANet). Finally, we combine the MDCNAResnet with a Bidirectional Gated Recurrent Unit (BiGRU) to further capture the contextual semantic information of EEG signals. Experimental results on the DEAP dataset demonstrate the effectiveness of our approach, achieving accuracies of 98.63% and 98.89% for Valence and Arousal dimensions, respectively.

## 1. Introduction

With increasing attention to emotional and affective cognition, emotion recognition plays a pivotal role in various domains, including psychological well-being, human-computer interaction, and personalized recommendations [1,2,3]. Traditional approaches often rely on psychological questionnaires or facial expression analysis to gather emotional information; however, these methods are susceptible to subjective factors and external environmental influences. In contrast, emotion recognition methods based on physiological signals, particularly EEG signals, offer greater objectivity and stability [4,5,6]. Nevertheless, current EEG-based emotion recognition methods still confront several challenges, such as the complexity of signals, individual variations, and noise interference [7,8,9,10]. Therefore, accurate and reliable recognition of EEG signals [11] contributes to improving the precision and credibility of systems. In recent years, researchers have extensively studied emotion recognition based on EEG signals using machine learning and deep learning methods [12,13,14].

In traditional machine learning algorithms, Seo et al. [15] extracted features from EEG signals including absolute band power, normalized absolute band power, differential entropy, differential asymmetry, and rational asymmetry. These features were then classified using algorithms such as Support Vector Machine (SVM), random forest, and K-Nearest Neighbors (KNN), achieving good classification results in emotion recognition. Mehmood et al. [16] employed Hjorth parameters to extract features from all EEG channels and used SVM and KNN as two different classification methods. However, machine learning-based emotion recognition algorithms heavily rely on manual feature extraction, which poses limitations on capturing and understanding complex emotional expressions.

In recent years, deep learning algorithms have been widely applied in various fields such as image and sequential signal processing [17,18]. Deep neural networks have the capability to automatically extract high-dimensional features from signals and demonstrate strong generalization and robustness, making them suitable for emotion recognition tasks. The Feature Pyramid Network (FPN) [19,20] is a common method used for multi-scale feature fusion. It constructs a feature pyramid with different resolutions and utilizes both top-down and bottom-up pathways to integrate features, thereby enhancing feature representation. FPN has been successfully applied in EEG signal research and achieved good classification performance [21]. On the other hand, Deformable Convolution (DCN) [22,23] is a type of convolution operation with deformation-awareness. By learning the local sampling positions and deformations of features, it can better adapt to geometric variations in input data. Recently, researchers have started incorporating deformable convolution into EEG signal processing tasks and achieved promising classification results [24]. Additionally, Zheng et al. [25] introduced the Deep Belief Network (DBN) to construct an emotion recognition model based on EEG. DBN trained with differential entropy features extracted from multi-channel EEG data to study key frequency bands and EEG channels in emotion recognition. Shen et al. [26] proposed a four-dimensional Convolutional Recurrent Neural Network (CNN-LSTM) that integrated frequency, spatial, and temporal information from multi-channel EEG signals. This method achieved accuracies of 94.22% and 94.58% on the DEAP dataset for Valence and Arousal dimensions, respectively. Zhang et al. [27] introduced a three-dimensional convolutional neural network (FSA-3D-CNN) that combined the frequency-space attention mechanism. It designed an adaptive mechanism to allocate weights to the frequency and spatial channels of EEG signals, mining the spatial and frequency information that more significantly reflected changes in emotional states. The accuracies of the DEAP dataset reached 95.87% and 94.58% for Valence and Arousal dimensions, respectively. In pursuit of achieving higher levels of performance, researchers have begun to explore the utilization of spatiotemporal deep neural networks. Li et al. [28] employed a bidirectional long short-term memory (BiLSTM) network to capture intrinsic spatial relationships within both intra- and inter-brain EEG electrode regions. They introduced a regional attention mechanism for learning, and this approach yielded promising results on the SEED dataset. Toma et al. [29] proposed a four-channel convolutional bidirectional long short-term memory (Bi-LSTM) network that effectively leverages spatiotemporal features from multiple channels collected from polysomnography (PSG) recordings for automatic sleep stage detection. This network demonstrated favorable outcomes on the Sleep EDF-1 dataset. Yuvaraj et al. [30] used a pre-trained 3D-CNN MobileNet model for transfer learning on EEG signals to extract features for emotion recognition.

These studies have introduced a series of innovative models that have achieved remarkable performance in emotion recognition tasks using EEG signals. They have contributed theoretical insights and approaches to feature extraction and the classification of EEG signals for emotion recognition, yielding promising results. However, existing EEG-based emotion recognition methods still face challenges in handling signal complexity, individual variations, noise interference, and inadequate multi-scale information fusion. Additionally, issues such as suboptimal feature extraction and insufficient multi-scale information fusion persist. Firstly, information loss is prone to occur during the mapping process from low-dimensional feature matrices to high-dimensional feature matrices. Secondly, there is insufficient extraction of time-frequency and spatial domain features, and traditional convolutional kernels are limited to square or rectangular shapes and cannot dynamically adapt to recognition targets. Finally, these studies often treat different electrode channels as independent entities, and the research on the correlation between electrode channels has not been sufficiently explored.

To overcome these limitations, this study introduces an innovative approach, termed Multi-scale Deformable Convolutional Interacting Attention Network based on Residual Network (MDCNAResnet). Leveraging the advantages of deep learning, our method is tailored to address the complexity and multi-scale nature of EEG-based emotion recognition tasks. By incorporating deformable convolutions and attention mechanisms, our aim is to enhance the model’s feature extraction capability, enabling it to capture subtle variations in emotional information more effectively. Furthermore, the incorporation of a bottom-up feature pyramid network facilitates the efficient fusion of multi-scale information, thereby further enhancing the accuracy of emotion recognition.

In summary, the main contributions of this paper are as follows:(1)Proposed a method called Multi-scale Deformable Convolutional Interacting Attention Network based on Residual Network (MDCNAResnet), specifically optimized for EEG emotion recognition tasks.(2)Introduced deformable convolutions and attention mechanisms to enhance feature extraction capabilities and focus on important information.(3)Implemented a bottom-up feature pyramid network to fuse multi-scale information, thereby improving the accuracy of emotion recognition(4)Conducted experiments on the well-established DEAP and SEED datasets, achieving significantly superior performance and validating the effectiveness of the proposed approach.

In the following sections, we provide a detailed exposition of the MDCNAResnet method and substantiate its superior performance in the field of emotion recognition through experimental results.

## 2. Materials and Methods

Figure 1 illustrates the framework of MDCNAResnet-BiGRU for emotion recognition based on EEG signals. It consists of three parts: constructing the three-dimensional feature matrix, the MDCNAResnet-BiGRU network model, and the classifier. In the following section, we provide a detailed explanation of each part in the following order.

### 2.1. 3D Feature Matrix

In order to generate the 3D feature matrix from the original multi-channel EEG signals, this study employs a sliding window approach with a window length of 1 s to segment the EEG signals from the last 60 s. The sliding window has a step size of 1 s, and there is no overlap between the segmented samples [31]. Each segment is assigned the same label as the original EEG signals. Subsequently, Butterworth bandpass filters are applied to divide each segment into several frequency bands, namely theta (4–8 Hz), alpha (8–14 Hz), beta (14–31 Hz), and gamma (31–45 Hz). From each frequency band, the differential entropy (DE) feature of the EEG channels is extracted, which has been proven to be the most stable feature for emotion recognition [32]. The differential entropy feature can differentiate the low-frequency and high-frequency energy of the EEG signals, which is the generalized form of Shannon information entropy on continuous variables. The calculation process is shown in Formula (1).
(1)$$DE=-\int_{X}p(x)\log(p(x))dx$$
where x represents the specific frequency band signal obtained from a particular EEG channel, p(x) is the probability density function of x, and X represents the range of possible values for the information. For a segment of EEG signal approximated as following a Gaussian distribution $N(\mu_{},\sigma_{i}^{2})$ [26], the calculation of the differential entropy is shown in Equation (2):(2)$$\begin{array}{ll} DE & =-\int_{a}^{b}\frac{1}{\sqrt{2\pi \sigma_{i}^{2}}}e^{-\frac{{\left(x-\mu \right)}^{2}}{2\sigma_{i}^{2}}}\log\left(\frac{1}{\sqrt{2\pi \sigma_{i}^{2}}}e^{-\frac{{\left(x-\mu \right)}^{2}}{2\sigma_{i}^{2}}}\right)dx\\ & =\frac{1}{2}\log\left(2\pi e\sigma_{i}^{2}\right) \end{array}$$

It is equal to the logarithm of the energy spectrum in a specific frequency band. Where e represents the Euler’s constant and σ represents the standard deviation of x.

Assuming an original EEG signal is represented as Sn∈ℝm×r, where m and r represent the number of electrodes and the sampling rate of the original EEG signal, respectively. For each group of EEG sample signals, we calculate the DE feature for each frequency band using a sliding window of 1 s. Therefore, the EEG segment is transformed into DE segments Dn∈ℝm×d, where d represents the number of frequency bands, which is set to 4 in this paper, as depicted in Figure 2.

Later, in order to utilize the spatial information of electrodes, we transform all EEG channels into a 2D grid map. In this study, we used 32-channel EEG data, with each of the 32 channels representing Fp1, AF3, F3, F7, FC5, FC1, C3, T7, CP5, CP1, P3, P7, PO3, O1, Oz, Pz, Fp2, AF4, Fz, F4, F8, FC6, FC2, Cz, C4, T8, CP6, CP2, P4, P8, PO4, and O2. Figure 3 displays the standardized flat map of the international 10/20 system and its corresponding 2D matrix. The left side of the figure represents the international 10/20 system, where solid circles indicate the EEG electrodes used in the DEAP dataset, and hollow circles represent the unused test points. In the feature matrix, the frequency domain features extracted from different EEG channels are placed in the corresponding positions of the matrix based on their relative spatial coordinates. The positions in the matrix that correspond to unused electrodes are set to 0. Therefore, the DE segment is transformed into Xn∈ℝh×w×d, where h and w represent the height and width of the 2D matrix, respectively. In this study, we set h = 9 and w = 9. Finally, the constructed 3D feature matrix is input into the neural network model.

### 2.2. Efficient Channel Attention Network

In this paper, ECANet is introduced into the network model. The principle of ECANet is shown in Figure 4. Without reducing the dimensionality, the input feature map χ undergoes global average pooling across all channels. ECA learns through a one-dimensional convolution with weight sharing and considers each channel’s interaction with its k neighbors to capture inter-channel interactions. In ECANet, the matrix $W_{k}$ is used to learn channel attention, as shown in Equation (3).
(3)$$W_{k}=\left[\begin{array}{cccccccc} w^{1,1} & \cdots & w^{1,k} & 0 & 0 & \cdots & \cdots & 0\\ 0 & w^{2,2} & \cdots & w^{2,k+1} & 0 & \cdots & \cdots & 0\\ \vdots & \vdots & \vdots & \vdots & \ddots & \vdots & \vdots & \vdots \\ 0 & \cdots & 0 & 0 & \cdots & w^{c,c-k+1} & \cdots & w^{c,c} \end{array}\right]$$

The weight vector $W_{k}$ is composed of *k* × **C* parameters, avoiding complete independence among different channels and considering the interactions between them.

In ECANet, the parameter *k* represents the size of the kernel in the one-dimensional convolution operation. It is proportional to the coverage range of cross-channel information interaction and the channel dimension *C*. Nonlinear functions are employed, and the number of convolution kernels is set to 2, raised to the power of *k* (since channel sizes are typically exponential multiples of 2). This relationship is expressed in Equation (4).
(4)$$C=\phi (k)=2^{\left(\lambda *k-b\right)}$$

In the equation, *k* represents the size of the kernel in one-dimensional convolution, *γ* is a constant used to calculate the parameter in the convolution size, and it is used to compute the parameter b in the convolution size. The specific calculation formula is as follows:(5)b=γ−1

The formula for calculating the convolution size is as follows:(6)$$\begin{array}{l} k=\psi (C)=|\frac{\log_{2}(C)+b}{\gamma }{|}_{\mathrm{odd}} \end{array}$$

According to the setting in reference [33], we have the following values: *γ* = 2, b = 1, *C* represents the number of network channels, $|t{|}_{\mathrm{odd}}$ denotes the nearest odd integer to *k*. If *k* is an even number, then $|t{|}_{\mathrm{odd}}$ is chosen as *k* + 1.

Mathematically, the ECA block can be described as follows:

Given an input feature map X∈ℝC×H×W, where C is the number of channels, H is the height, and W is the width, the ECA block computes the output feature map Y∈ℝC×H×W as:(7)Yi=Xi·σ1H×W∑j=1H×WXij+Xi
where: i represents the channel index (1≤i≤C), j iterates through all spatial locations (1≤j≤H×W), Xi is the ith channel of the input feature map, σ is the sigmoid activation function, H and W are the height and width of the feature map, respectively.

In this equation, the term $\frac{1}{H\times W}\sum_{j=1}^{H\times W}X_{ij}$ computes the average activation of the ith channel across all spatial locations. The sigmoid activation function σ introduces a gating mechanism that modulates the importance of each channel. Channels with higher average activations are emphasized, while channels with lower average activations are suppressed.

The ECA_block is embedded into the residual structure, resulting in the architecture shown in Figure 5.

### 2.3. Deformable Convolutional Network

A Deformable Convolutional Network (DCN) is a convolutional operation proposed by Dai et al. [34] in 2017. It introduces additional offset variables at each sampling point, which allows the convolutional kernel to effectively sample the data without being restricted to a fixed shape. This significantly increases the receptive field of the convolutional operation and enhances the modeling capability of convolutional neural networks for irregular objects. The operation of a DCN is illustrated in Figure 6. Initially, a DCN performs convolutional operations on the input feature map to obtain a set of predicted offset maps. The size of the offset feature map remains the same as the input feature map. The channel dimension of the offset feature map is 2N, where 2 represents the two values (*x*, *y*) for each offset and N is the number of pixels in the convolutional kernel.

In this study, the model utilizes convolutional kernels with a size of 3 × 3. The parameter *R* is defined as the sampling region of the convolutional kernel.
(8)$$R=\left\{\left(-1,-1\right),(-1,0),\ldots (0,1),(1,1)\right\}$$

For traditional convolutional operations, the calculation for each position on the output feature map y is as follows:(9)$$y(p_{0})=\sum_{P_{n}\in R}w(p_{n})\cdot x(p_{0}+p_{n})$$
where $p_{n}$ represents the offset at each position of the convolutional kernel corresponding to $p_{0}$.

For deformable convolution operation, the calculation for each position $p_{0}$ on the output feature map y is as follows:(10)$$y(p_{0})=\sum_{P_{n}\in R}w(p_{n})\cdot x(p_{0}+p_{n}+\Delta p_{n})$$
where $\Delta p_{n}$ represents the learned offset values. Now, the sampling is performed at irregular offset positions $p_{n}+\Delta p_{n}$.

The deformable convolution operation, performed on the input feature map, overcomes the fixed patterns of the traditional convolution operation by introducing offset values and different weights that modify the receptive field. This significantly increases the range of the receptive field and allows it to better converge on the representation region, enhancing the network’s adaptive capability to capture features without redundant information. When the original EEG data are transformed into a three-dimensional feature matrix, more comprehensive and accurate features can be extracted from the EEG signals.

The DCN module is embedded into the ECA_block, resulting in the DCNA_block, as shown in Figure 7.

The DCNA-ResNet18 network, as shown in Figure 8, is an improved version of ResNet18, by replacing the traditional convolutional part with the deformable convolutional operation and embedding the ECANet module. This network combines the benefits of deformable convolution and the channel attention mechanism, enhancing the modeling capability and feature representation ability for emotion recognition from EEG signals.

### 2.4. Bottom-Up Feature Pyramid Network

A Traditional Feature Pyramid Network (FPN) utilizes the top-down pathway to fuse positional information from the lower-level feature maps. However, the upsampling operation in the top-down pathway may lead to information loss, especially when upsampling low-resolution feature maps. On the other hand, the bottom-up downsampling operation helps to preserve more original feature information, improving the quality and detail preservation of the feature maps. Additionally, the bottom-up downsampling operation increases the effective receptive field of small-scale feature maps, allowing them to capture more local details. By fusing features from different scales, the FPN enables small-scale feature maps to better express local features and subtle variations.

The structure of the BU-FPN is illustrated in Figure 9. Taking ResNet18 as the backbone, C2 to C5 feature maps are obtained through convolutional operations. Then, a 1 × 1 convolutional layer is applied to maintain the consistency of feature map dimensions, resulting in T2 to T5 feature maps. The downsampling process starts from T2 and continues as follows:(11)$$m=\frac{n+2p-f}{s}+1$$
whereas m represents the size of the output feature map D, *n* represents the size of the input feature map T, *p* denotes padding, which is the number of pixels added around the input, *f* represents the filter size or the size of the convolutional kernel, and *s* represents the stride, which is the sliding step size. After downsampling T2, it is added to T3 to obtain the tensor D3. The same process is applied to D3, which is then added to T4 to obtain D4. This process continues until the feature map D5 is obtained. Finally, a 3 × 3 convolution is applied to D5, resulting in a smaller-sized feature map P5 that contains more rich information.

### 2.5. Bidirectional Gated Recurrent Unit

RNN (Recurrent Neural Network) networks are effective in handling time-series signals. However, simple RNNs suffer from the problems of vanishing and exploding gradients, making them difficult to apply to tasks with long-term dependencies. To address these issues, Hochreiter [35] and Chung [36] proposed LSTM (Long Short-Term Memory) networks and GRU (Gated Recurrent Unit) networks, respectively. LSTM networks track long-term information through the use of gates, while GRU networks achieve similar effectiveness as LSTM while having a simpler structure and lower computational complexity.

A BiGRU (Bidirectional Gated Recurrent Unit) consists of two regular GRU units that process two different directions of the time sequence (forward and backward). The outputs of the two sequences are then concatenated. In this way, BiGRU is able to capture both past and future context information. In the context of emotion recognition, individuals often generate corresponding feedback in the brain based on the intensity of the stimulus they receive. Therefore, in this study, BiGRU is employed to extract contextual semantic information from both forward and backward sequences. The principle of BiGRU is illustrated in Figure 10.

The proposed network architecture, as depicted in Figure 11, begins with the preprocessing of multi-channel EEG signals. Employing a sliding window approach, the raw EEG signals are segmented, with each segment assigned the same label as the original EEG signal. These segments are then divided into four frequency bands—θ (4–8 Hz), α (8–14 Hz), β (14–31 Hz), and γ (31–45 Hz)—using Butterworth bandpass filters. From each frequency band, Differential Entropy (DE) features of EEG channels are extracted, differentiating the low- and high-frequency energy of EEG signals and constructing a three-dimensional feature matrix based on the relative positions of EEG signals.

Subsequently, the Multi-scale Deformable Convolutional Interacting Attention Network based on Residual Network (MDCNAResnet) is introduced. In this network, the Efficient Channel Attention (ECA) mechanism is initially employed to capture crucial inter-channel information. This information is then combined with deformable convolutions to acquire richer spatial features. The deformable convolution operations facilitate the capture of nonlinear variations within EEG signals and enable the adaptive sampling of convolutional kernels based on spatial distribution. Additionally, the Bidirectional Gated Recurrent Unit (BiGRU) is integrated into the network to extract contextual semantic information from both forward and backward directions. This integration achieves the fusion of multi-scale spatial features and contextual semantic features.

Toward the top of the network, a Bottom-Up Feature Pyramid Network (BU-FPN) is applied to further fuse multi-scale features. This network, through a bottom-up downsampling process, progressively extracts and integrates features of different scales, enhancing the model’s ability to capture both fine-grained details and holistic information. Ultimately, emotion categories are predicted through fully connected and softmax classification layers.

The interplay between different components within our method is pivotal. The extraction of DE features enriches subsequent attention mechanisms and deformable convolution operations. The ECA mechanism enhances inter-channel interactions, feeding into deformable convolutions. The BiGRU network captures the temporal information of EEG signals and, when combined with spatial features, facilitates global contextual modeling. Lastly, BU-FPN further fuses multi-scale features, leading to improved accuracy in emotion recognition.

## 3. Experimental Results and Analysis

### 3.1. Dataset and Data Preprocessing

#### 3.1.1. Dataset Introduction

The DEAP dataset (Koelstra et al. [37]) consists of physiological signals recorded from 32 participants while they watched 40 min of music videos. Each music video lasted for 1 min, and the dataset includes the participants’ physiological signals as well as their ratings of Valence, Arousal, Dominance, and Liking using psychological scales. Facial expression videos are also available for the first 22 participants, as shown in Table 1. The emotional categories of each video segment are labeled based on graded ratings of Arousal and Valence (ranging from 1 to 9). We use a threshold of 5 to divide the labels into two binary classification problems.

The dataset includes 32 participants, including 16 males and 16 females, with an average age of 26.9 years. Their EEG signals were recorded using a Biosemi ActiveTwo device with 32 channels, following the international 10–20 system, at a sampling frequency of 512 Hz. Before the dataset release, the Electromyography and Electrooculography signals were removed. The EEG signals were downsampled to 128 Hz. They were passed through a bandpass filter from 4 to 45 Hz to remove noise. Each trial contains 63 s of EEG signals, with the first 3 s representing the pre-baseline signal in a relaxed state. The remaining 60 s of EEG recordings capture the emotional signals.

In addition to the DEAP dataset, the SEED dataset [38] was utilized for validating the model’s performance. The SEED dataset encompasses data representing three distinct emotional categories: positive, neutral, and negative. Fifteen healthy participants were involved in the acquisition of EEG signals, with each participant undergoing three experimental sessions. Each session consisted of observing fifteen segments, each comprising four stages: a 5-s initial prompt, a 4-min clip period, a 45-s self-assessment, and a 15-s rest interval. Similar to the DEAP dataset, the SEED dataset underwent training and validation using the same procedures.

We conducted experiments on the well-established DEAP and SEED datasets to evaluate the performance of our proposed approach. The choice of these datasets was driven by specific motivations. The DEAP dataset, known for its wide use in emotion recognition research, offers diverse emotional stimuli and rich emotional information. This enables us to comprehensively explore the challenges of emotion recognition tasks. On the other hand, the SEED dataset provides a different context with distinct emotional stimuli and experimental conditions, allowing us to assess the generalization capability of our method across varying contexts.

#### 3.1.2. Data Preprocessing

In the DEAP dataset, the raw EEG signals are represented as 32 (subjects) × 40 (trials) × 40 (channels) × 8064 (samples), where 8064 represents 128 (samples) × 63 (s). The labels are represented as 40 (trials) × 4. To preprocess the raw data, the required 32 EEG channels are extracted from the 40 channels. Considering the delay in human visual response, the first 3 s are taken as the baseline, and the EEG signals from the subsequent 60 s are extracted as the experimental data. The preprocessed data is represented as 32 (subjects) × 40 (trials) × 32 (channels) × 7680 (samples). The labels are selected based on Valence and Arousal dimensions, resulting in a size of 40 (trials) × 2.

The EEG sequences are segmented into non-overlapping 1-s segments, resulting in 60 segments per trial. Each segment contains 128 sampling points, and each sampling point includes 32 channels. Thus, the EEG data for each subject can be represented as 40 × 128 × 60 × 32, and the data are transformed to a dimension of 2400 × 32 × 128, where each subject has 2400 EEG segments, and each segment has a size of 32 × 128. The labels undergo the same dimensional transformation and can be represented as 2400 × 1.

Then, differential entropy features are extracted from the four frequency bands of the original features. Then, the data from the 32 channels are transformed into a two-dimensional mesh structure, resulting in a representation of 128 × 2400 × 9 × 9. After concatenating the four features, a three-dimensional feature matrix is obtained with dimensions of 307,200 × 9 × 9 × 4. This means that the deep model is trained on 307,200 samples, with corresponding labels of size 307,200 × 1. The SEED dataset underwent preprocessing using the same procedures as the DEAP dataset.

#### 3.1.3. Experimental Setup

The MDCNAResnet-BiGRU Network is trained with a batch size of 128 using the Adam optimizer with a learning rate of 0.001. The training is conducted for 100 epochs. The model is implemented in TensorFlow and trained on an NVIDIA GTX 1650 GPU. The performance of MDCNAResnet-BiGRU was evaluated using the following metrics: Accuracy, Precision, Standard Deviation, and Recall. The specific calculation formulas are as follows:(12)$$\mathrm{Accuracy}=\frac{\mathrm{TP}+\mathrm{TN}}{\mathrm{TP}+\mathrm{FN}+\mathrm{FP}+\mathrm{TN}}\times 100\%$$
(13)$$\mathrm{Precision}=\frac{\mathrm{TP}}{\mathrm{TP}+\mathrm{FP}}\times 100\%$$
(14)$$\mathrm{Recall}=\frac{\mathrm{TP}}{\mathrm{TP}+\mathrm{FN}}\times 100\%$$
(15)$$F-\mathrm{score}=2\times \frac{\mathrm{Precision}\times \mathrm{Recall}}{\mathrm{Precision}+\mathrm{Recall}}\times 100\%$$

In the context of classification, TP (True Positives) represents the number of instances that are correctly classified as positive, while TN (True Negatives) represents the number of instances that are correctly classified as negative. FP (False Positives) represents the number of instances that are incorrectly classified as positive, and FN (False Negatives) represents the number of instances that are incorrectly classified as negative.

The proposed MDCNAResnet-BiGRU network takes a 3D feature matrix of size 9 × 9 × 4 as input. In this study, four frequency bands are selected for combination, as previous research has shown that the combination of all bands can complement each other and yield better results compared to individual bands (Zheng and Lu, 2015 [25]). Different parameter settings can have an impact on the performance of neural networks. Therefore, we investigate the impact on performance by changing the value of K in K-fold cross-validation and the position of the deformable convolutional layer. Subsequently, the overall performance of the proposed model is evaluated. Finally, comparisons are made with other emotion recognition algorithms.

### 3.2. The Effect of K-Values

In k-fold cross-validation, different values of k can impact the final classification accuracy. By varying the size of k, performance comparisons were conducted on the DEAP dataset. The experimental results are shown in Figure 12.

Based on the above analysis, we found that at K = 5, compared to other K values, there was an improvement of 2.63–4.80% in the Valence dimension and 2.63–4.69% in the Arousal dimension. Therefore, we chose five-fold cross-validation. For the DEAP dataset, according to the experimental results, each subject’s samples are randomly divided into five groups, with one group used as the test set and the remaining four groups used as the training set.

### 3.3. The Effect of DCNA-ResNet18

Different network architectures can affect the final classification accuracy. By changing the position of the deformable convolutional layer in the ResNet18 network, we compared the performance of the DEAP dataset. The experimental results are shown in Table 2. The numbers in parentheses indicate the position of the deformable convolution in the residual structure. For example, (4) indicates replacement in the fourth residual structure.

Based on the above analysis, we found that replacing the deformable convolutional layer in the fourth residual structure leads to improved performance compared to other replacement strategies. In the Valence dimension, there was an improvement of 1.29–3.35%, and in the Arousal dimension, there was an improvement of 1.17–3.25%. Therefore, we chose to replace the deformable convolutional layer in the fourth residual structure.

### 3.4. Ablation Experiments

Ablation experiments were conducted to validate the effectiveness of the proposed model. The baseline model, referred to as Resnet-BiGRU, was compared against ECAResnet-BiGRU, where the efficient channel attention mechanism was introduced in Resnet-BiGRU. Additionally, the traditional convolution in the fourth residual structure of Resnet was replaced with deformable convolution, forming DCNAResnet-BiGRU. Finally, BU-FPN was employed for multi-scale fusion, resulting in MDCNAResnet-BiGRU.

Table 3 presents the average performance metrics for all participants. It can be observed that Resnet-BiGRU achieved relatively low recognition accuracy. However, ECAResnet-BiGRU exhibited improved accuracy in both Valence and Arousal dimensions, indicating the effectiveness of the channel attention mechanism. By replacing the traditional convolution with deformable convolution in DCNAResnet-BiGRU, the recognition accuracy increased by 3.53% in the Valence dimension and 3.85% in the Arousal dimension, demonstrating the capability of deformable convolution in extracting hidden features and enhancing network performance. Finally, with the adoption of BU-FPN for multi-scale fusion, MDCNAResnet-BiGRU further extracted multi-scale features, achieving recognition accuracies of 98.63% and 98.89% in the two dimensions, respectively. The results of the ablation experiments confirm the effectiveness of the proposed model in EEG-based emotion recognition. Figure 13 illustrates the recognition accuracy of the MDCNAResnet-BiGRU model for each participant in the Valence and Arousal dimensions.

### 3.5. MDCNAResnet-BiGRU Performance Testing on Different Datasets

In addition to the DEAP dataset, the performance of MDCNAResnet-BiGRU was also validated on the SEED dataset. As shown in Table 4, accuracy, precision, and recall were computed. The results demonstrate that the model performs remarkably well on the SEED dataset as well, achieving an accuracy of 98.13% in classifying emotions into three categories. Thus, MDCNAResnet-BiGRU exhibits outstanding performance not only on the DEAP dataset but also on the SEED dataset.

### 3.6. Leave-One-Subject-Out (LOSO) Experiment

To objectively assess the performance of our model across different individuals, we employed the leave-one-subject-out (LOSO) cross-validation strategy. In this approach, each participant’s data was used as the test set, while the data from all other participants were utilized as the training set, and this process was repeated for each participant. Table 5 summarizes the average recognition accuracy of MDCNAResnet-BiGRU on the SEED dataset. To validate the effectiveness of MDCNAResnet-BiGRU, we conducted comparisons with other deep learning methods. Li et al. [39] introduced a novel bi-hemispheric discrepancy model (BiHDM) for emotion recognition, aiming to simulate the asymmetric differences between EEG signals. Li et al. [40] proposed a spatial-temporal graph attention network with a transformer encoder (STGATE), a novel graph neural network, to learn graph representations of EEG signals and enhance emotion recognition performance. Zhu et al. [41] introduced the concept of energy sequences to mitigate noise overlay caused by feature analysis and extraction, and they proposed a deep network model based on dynamic energy features using bidirectional long short-term memory (Bi-LSTM). Li et al. [42] proposed a self-organizing graph neural network (SOGNN) for cross-subject EEG-based emotion recognition, where the graph structure of SOGNN is dynamically constructed by self-organizing modules for each signal. Topic et al. [43] presented a novel emotion recognition model that creates feature maps based on EEG signal features using topography (TOPO-FM) and holography (HOLO-FM) representations:

Following the validation using the leave-one-subject-out (LOSO) approach, we obtained the following results: In the emotion recognition task, our proposed network model achieved an average accuracy of 90.47% under leave-one-subject-out (LOSO) validation. These results substantiate the robustness and generalization capability of our model.

### 3.7. Algorithm Comparison

The proposed model was compared with other recent network models, and the results are shown in Table 6:

(1) 3DCNER (Zheng et al., 2021 [44]): This method utilizes a three-dimensional feature matrix and a neural network for emotion recognition. It extracts features from each EEG segment to construct a three-dimensional feature matrix, and then employs the CNNFF network framework for emotion recognition.

(2) FSA-3D-CNN (Zhang et al., 2022 [27]): This method introduces a four-dimensional feature matrix that incorporates the temporal and spatial information of EEG signals. It applies 3D-CNN for emotion classification.

(3) 4D-CRNN (Shen et al., 2020 [26]): This approach constructs a four-dimensional feature matrix and employs a convolutional recurrent neural network (CRNN) to extract spatial and temporal features for EEG emotion recognition.

(4) CNN-LSTM (Chen et al., 2020 [45]): This method represents the temporal and spatial correlations using raw EEG signals and utilizes a hybrid CNN and LSTM architecture, both in cascade and parallel, to predict the emotion classification of each EEG sample.

(5) ACRNN (Tao et al., 2020 [46]): This approach introduces an attention-based hybrid CNN and RNN network, where the CNN is used to extract spatial features from EEG signals weighted by the attention module, and the RNN combines the temporal features for emotion recognition.

(6) TSFFN (Sun et al., 2022 [47]): This approach introduces a multi-channel EEG emotion recognition model based on parallel transformers and a three-dimensional convolutional neural network (3D-CNN). It involves creating parallel channel EEG data and reconstructing EEG sequence data with positional information. The model then utilizes transformers and 3D-CNN to extract temporal and spatial features from EEG data.

(7) 3DCNN (Chao et al., 2020 [48]): This method employs an advanced convolutional neural network (CNN) design that consists of single-variable convolutional layers and multi-variable convolutional layers to process 3D feature matrices for emotion recognition.

(8) Multi-aCRNN (Xin et al., 2023 [49]): The multi-aCRNN model proposes a multi-view feature fusion attention convolutional recurrent neural network approach. It combines CNN, GRU, and attention mechanisms while employing label smoothing to reduce noise interference.

(9) RA2-3DCNN (Cui et al., 2022 [50]): This method presents an emotion recognition approach based on both two-dimensional convolutional neural networks and three-dimensional convolutional neural networks.

(10) 4D-aNN (Xiao et al., 2022 [51]): The 4D-aNN approach introduces adaptive allocation of weights to different brain regions and frequency bands using spectrum and spatial attention mechanisms. It utilizes a convolutional neural network (CNN) to process spectral and spatial information represented in 4D, combining both frequency and spatial dimensions.

(11) FCA-ReliefF, SVM (Pan et al., 2022 [52]): This paper proposes a two-stage feature optimization approach based on Feature Correlation Analysis (FCA) and the ReliefF algorithm to select key features, aiming to reduce feature dimensionality and enhance the accuracy of emotion recognition.

Our research aims to develop an efficient EEG emotion recognition system by integrating the Multi-scale Deformable Convolutional Interacting Attention Network based on Residual Network (MDCNAResnet) and attention mechanisms, along with the application of the Bottom-Up Feature Pyramid Network (BU-FPN). Through this approach, we achieved a series of significant accomplishments.

Firstly, our method holds scientific value in the field of EEG emotion recognition. We introduced the innovative Multi-scale Deformable Convolution operation within the network structure, which better captures spatial information within EEG signals and enhances emotion recognition performance. Additionally, the incorporation of the Efficient Channel Attention mechanism further intensifies the network’s focus on crucial information. Our research is not only methodologically innovative but has also demonstrated exceptional recognition accuracy in EEG emotion recognition tasks, significantly contributing to the field’s development.

Secondly, a comparison with existing methods highlights the substantial advantages of our approach. We conducted comprehensive performance comparisons with ten recent similar publications using the DEAP dataset. As indicated in Table 6, MDCNAResnet excels in emotion recognition tasks, significantly outperforming other methods. Our approach achieves recognition accuracies of 98.63% and 98.89% on the DEAP dataset and 98.13% on the SEED dataset, respectively, which far surpasses other methods. This further solidifies the remarkable advantages of our method in EEG emotion recognition.

In summary, our research is not only methodologically innovative but has also achieved significant accomplishments, contributing to the field with scientific value. The proposed MDCNAResnet method performs exceptionally well in EEG emotion recognition tasks, offering robust support for the advancement and application of emotion recognition technology.

## 4. Discussion

In this section, we delve into a detailed discussion of our proposed method, emphasizing the significance of key components, including deformable convolution, the attention mechanism, and the Bottom-Up Feature Pyramid Network (BU-FPN). The integration of these modules stands as a pivotal factor contributing to the success of our approach.

To begin, we adopt deformable convolution as the core element for feature extraction. Compared to traditional convolution, deformable convolution offers a flexible adaptation to feature patterns within varying receptive fields, thereby better capturing subtle variations within EEG signals. This capability is crucial for emotion recognition tasks, as the expression of emotional states may change across different frequency ranges and temporal scales. By introducing deformable convolution, our model excels in capturing these intricate patterns and changes, consequently enhancing the accuracy of emotion recognition.

Furthermore, we introduce the attention mechanism to enhance feature representation. The attention mechanism dynamically adjusts the weights of features, enabling the model to focus more on pertinent information related to emotional states. In our model, we employ the Efficient Channel Attention Network (ECANet) to optimize feature extraction, ensuring a stronger emphasis on meaningful features relevant to emotion recognition. This mechanism further amplifies the expressiveness of our model, aiding in more accurately capturing emotion-related information within EEG signals.

Lastly, we leverage the Bottom-Up Feature Pyramid Network (BU-FPN) for multi-scale feature fusion. Given that emotional expression may vary across different spatial scales, fusing features from different scales is paramount for enhancing emotion recognition accuracy. BU-FPN amalgamates features across different layers, enabling the model to simultaneously focus on local details and global context. This multi-scale feature fusion enhances our model’s comprehensive understanding and interpretation of emotion-related information within EEG signals, consequently boosting emotion recognition performance.

In summary, our approach significantly enhances EEG-based emotion recognition by introducing deformable convolution, the attention mechanism, and the Bottom-Up Feature Pyramid Network. These critical components play vital roles in capturing complex emotional expression patterns, optimizing feature extraction, and achieving multi-scale feature fusion. We firmly believe that the integration of these components, along with the remarkable achievements in emotion recognition tasks, offers new insights and possibilities for research and applications in this field.

## 5. Conclusions

This paper proposes a Multi-Scale Deformable Convolutional Network with Attention (MDCNAResnet), based on the residual network. The proposed model aims to extract emotion-related features from EEG signals. Firstly, we extract differential entropy features from the EEG signals of different channels and construct a three-dimensional feature matrix based on the relative positions of the electrodes. Then, we utilize multi-scale deformable interaction attention convolution and BiGRU to extract multi-scale spatial features and contextual semantic information from the EEG signals, achieving the fusion of spatial and temporal features. Experimental results on the DEAP dataset demonstrate that the proposed network model effectively extracts emotion-related features from EEG signals, achieving recognition accuracies of 98.63% and 98.89% in the Valence and Arousal dimensions and 98.13% on the SEED dataset, respectively. In future work, we aim to further simplify the network structure, reduce the number of parameters, shorten the model’s execution time, and improve the efficiency of EEG signal recognition. Additionally, we plan to incorporate multi-modal data to reduce the impact of individual differences on the network model’s performance and enhance its generalization ability.

## Figures and Tables

**Figure 1 brainsci-13-01293-f001:**
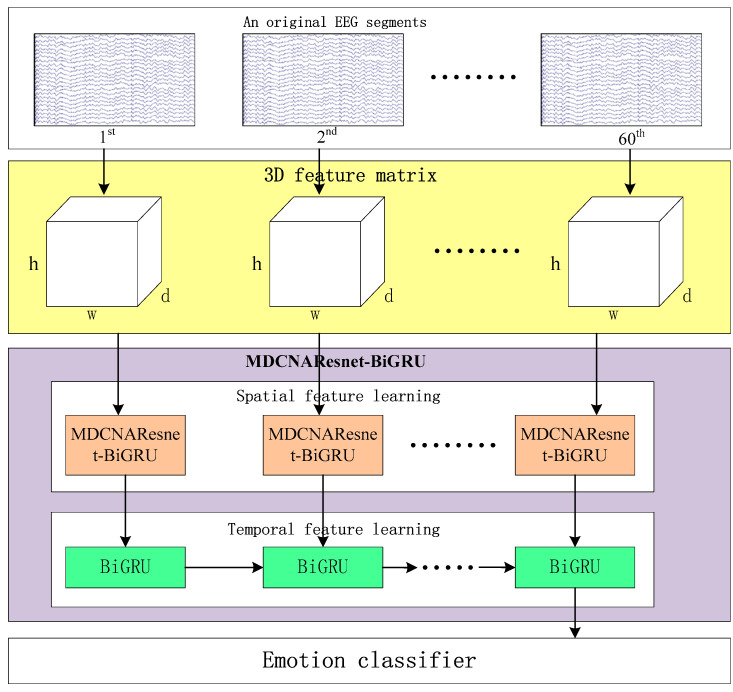
MDCNAResnet-BiGRU framework based on EEG signal emotion recognition.

**Figure 2 brainsci-13-01293-f002:**
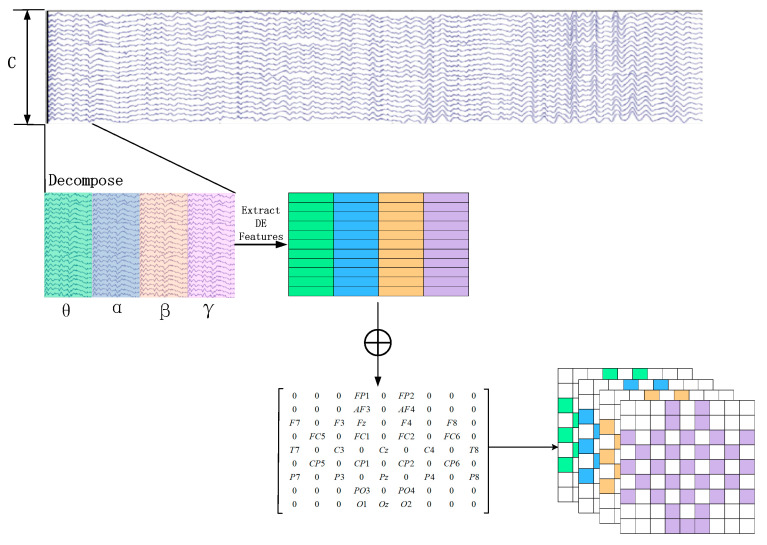
Construction of 3D Feature Matrix.

**Figure 3 brainsci-13-01293-f003:**
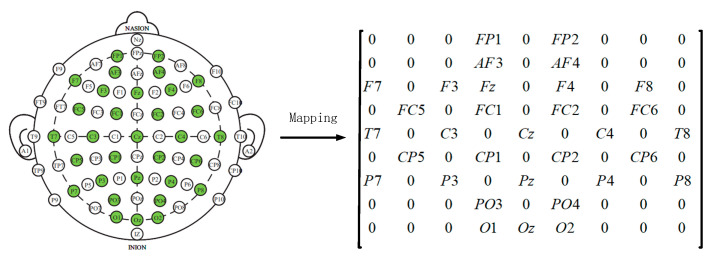
Plan and 2D Matrix of International 10/20 System Standards.

**Figure 4 brainsci-13-01293-f004:**
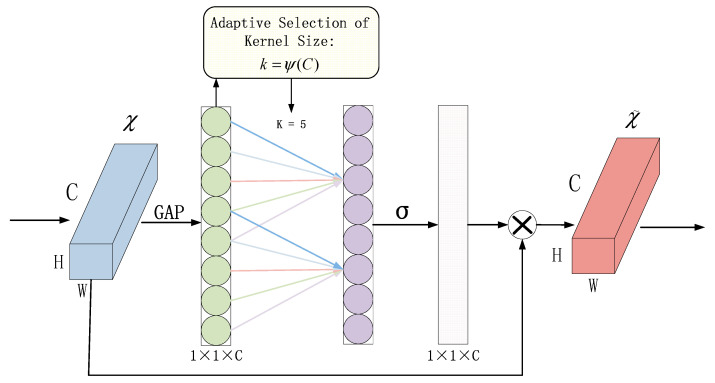
Structure diagram of efficient channel attention module.

**Figure 5 brainsci-13-01293-f005:**
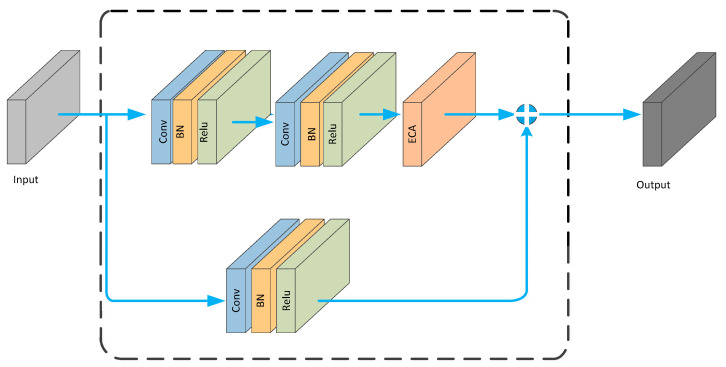
ECA_ Block structure diagram.

**Figure 6 brainsci-13-01293-f006:**
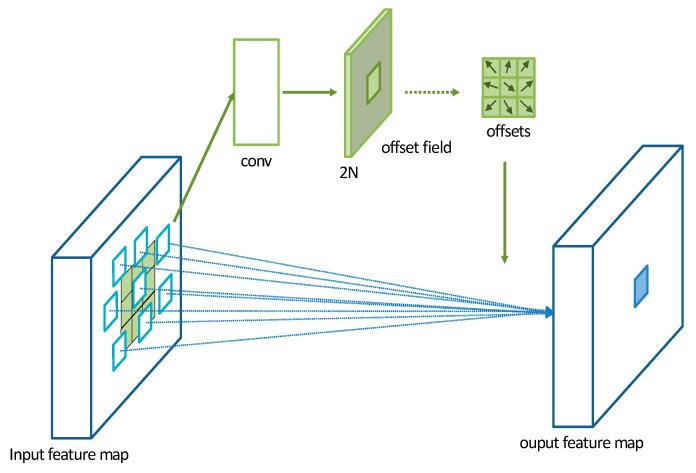
Schematic diagram of deformable convolution operation mechanism.

**Figure 7 brainsci-13-01293-f007:**
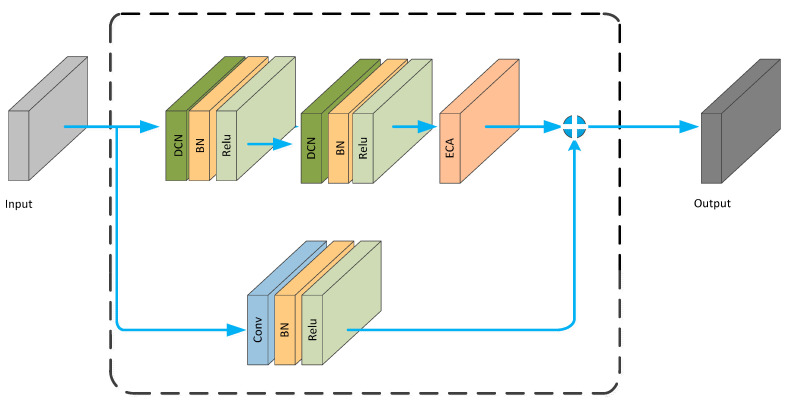
DCNA_ Block structure diagram.

**Figure 8 brainsci-13-01293-f008:**
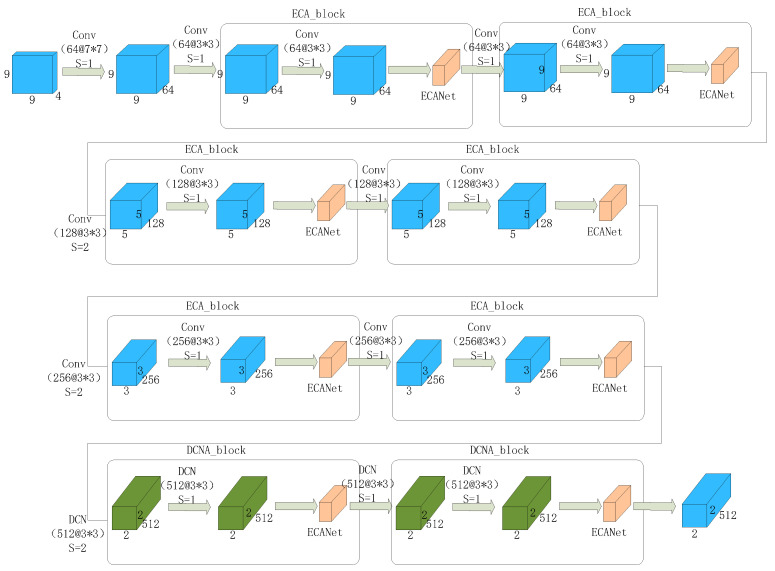
DCNA-ResNet18 Network Flowchart.

**Figure 9 brainsci-13-01293-f009:**
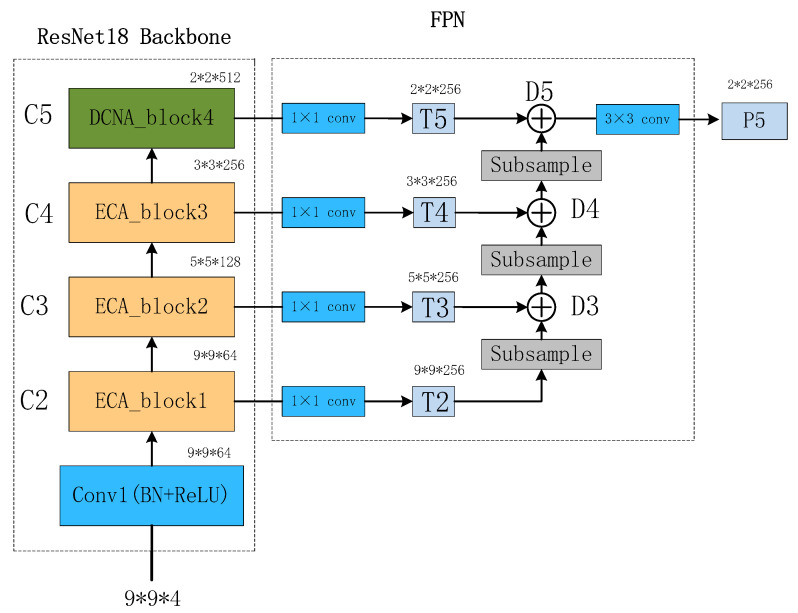
Bottom-up FPN structure diagram.

**Figure 10 brainsci-13-01293-f010:**
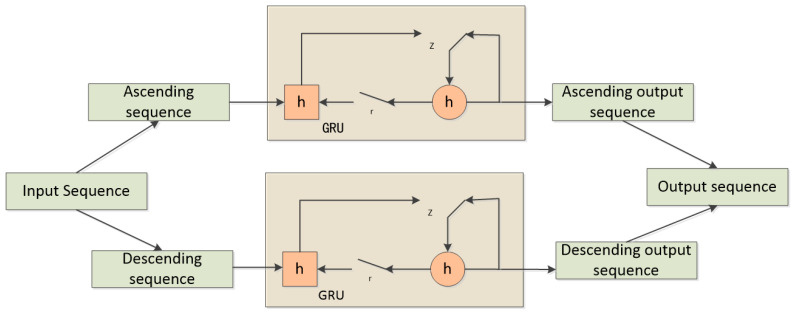
BiGRU schematic diagram.

**Figure 11 brainsci-13-01293-f011:**
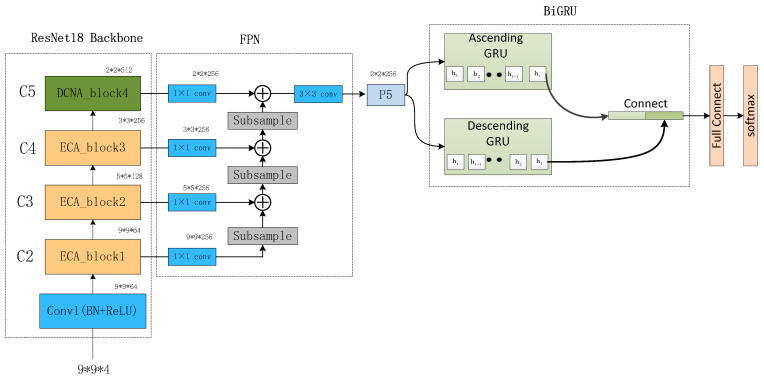
MDCNAResnet-BiGRU Network Structure Diagram.

**Figure 12 brainsci-13-01293-f012:**
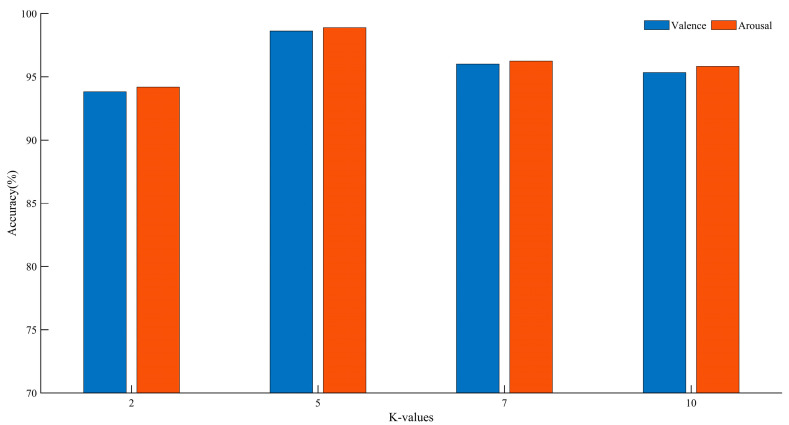
Recognition accuracy of different K values.

**Figure 13 brainsci-13-01293-f013:**
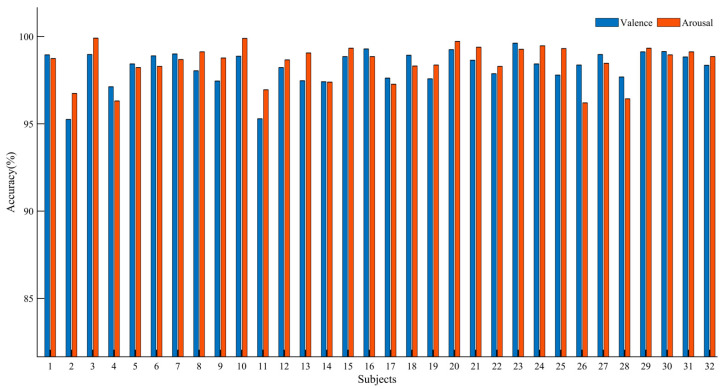
Overall Performance of MDCNAResnet-BiGRU on DEAP Datasets.

**Table 1 brainsci-13-01293-t001:** DEAP Dataset Format.

Data Name	Data Format	Data Content
Signal Data	40 × 40 × 8064	Video/Channel/Signal Length
Labels	40 × 4	Video/Labels

**Table 2 brainsci-13-01293-t002:** Identification accuracy of different network structures.

Method	DEAP-Valence	DEAP-Arousal
DCNA-ResNet18 (1)	95.28 ± 3.44%	95.62 ± 4.09%
DCNA-ResNet18 (2)	94.68 ± 2.82%	94.94 ± 3.63%
DCNA-ResNet18 (3)	93.22 ± 3.97%	93.64 ± 3.81%
DCNA-ResNet18 (4)	96.57 ± 2.67%	96.79 ± 3.06%

**Table 3 brainsci-13-01293-t003:** The ablation experiment of MDCNAResnet-BiGRU on the DEAP dataset.

Method	DEAP-Valence	DEAP-Arousal
Resnet-BiGRU	89.65 ± 3.93%	89.94 ± 4.26%
ECAResnet-BiGRU	93.04 ± 3.37%	92.94 ± 4.03%
DCNAResnet-BiGRU	96.57 ± 2.67%	96.79 ± 3.06%
MDCNAResnet-BiGRU	98.63 ± 1.51%	98.89 ± 1.22%

**Table 4 brainsci-13-01293-t004:** MDCNAResnet-BiGRU Performance Testing on Different Datasets.

Index	DEAP		SEED
	Valence	Arousal	
Accuracy	98.63%	98.89%	98.13%
Precision	98.70%	99.23%	98.53%
Recall	98.76%	99.17%	98.46%
F-score	98.72%	99.19%	98.49%

**Table 5 brainsci-13-01293-t005:** Leave-one-subject-out emotion recognition on the SEED dataset.

Model	SEED
BiHDM	85.40%
STGATE	90.37%
Bi-LSTM	89.42%
SOGNN	86.81%
Holo+FM/CNN+SVM	88.45%
Our method	90.47%

**Table 6 brainsci-13-01293-t006:** Performance comparison of different approaches.

Model	Signal	DEAP-Valence	DEAP-Arousal	SEED
3DCNER	EEG	93.61%	94.04%	—
FSA-3D-CNN	EEG	95.23%	95.87%	—
4D-CRNN	EEG	94.22%	94.58%	94.47%
CNN-LSTM	EEG	93.64%	93.26%	—
ACRNN	EEG	92.56%	93.81%	—
TSFFN	EEG	98.27%	98.53%	97.64%
3DCNN	EEG	96.45%	97.34%	—
multi-aCRNN	EEG	96.30%	96.43%	—
RA2-3DCNN	EEG	97.58%	97.19%	—
4D-aNN	EEG	96.90%	97.39%	96.25%
FCA-ReliefF, SVM	BVP, GSP, RSP, EMG	92.34%		—
Our method	EEG	98.63%	98.89%	98.13%

## Data Availability

The processed data required to reproduce these findings cannot be shared as the data also forms part of an ongoing study.

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
