# Peer review of "EEG Emotion Recognition by Fusion of Multi-Scale Features"

_brainsci, 2023, doi:10.3390/brainsci13091293_

Round 1
Reviewer 1 Report
An EEG-based emotion recognition method called MDCNAResnet is presented in the paper. Feature extraction and mapping challenges are addressed using deformable convolution and a Bottom-Up Feature Pyramid Network with Efficient Channel Attention. On the DEAP dataset, the method achieves the accuracy of 98.63% and 98.89% for Valence and Arousal dimensions, respectively.
This study has a lot of valuable information, and I believe it is comprehensive. The subject fascinates me. There are several significant corrections that need to be made to improve the scientific level of the article.
1. Compare the accuracy of the proposed system to the papers that proposed a spatial-temporal deep neural network for emotion recognition. In this case, it would be helpful to compare the accuracy to 10 papers that have been published in the last 3 years.
2- Authors should demonstrate why this network outperformed other networks by plotting the brain topography of each EEG channel's weight.
3- Comparing the proposed system with similar ones would be helpful. It is not clear whether the article is scientifically valuable or novel.
4- Describe the proposed approach in greater detail, paying particular attention to the relationships between its components, since these are the key components of the solution.
5- There should also be a report on the accuracy of the proposed network with the leave-one-subject-out validation method. Recently, most articles focus on emotion recognition with the leave-one-subject-out method.
6- Explain why what has been learned is valuable and exciting in a better way.
7- The "Introduction" section should be revised to include an accurate and informative literature review of existing methods and how the proposed method differs from them. Furthermore, the motivation and contribution should be clarified.
8- In the introduction, logic needs improvement. For example, the significance and reasons for using spatial-temporal deep neural networks for cognitive tasks. It would also be helpful to mention current progress and critical issues. Authors can use these articles to edit this section. From regional to global brain: A novel hierarchical spatial-temporal neural network model for EEG emotion recognition/ An End-to-End Multi-Channel Convolutional Bi-LSTM Network for Automatic Sleep Stage Detection/ Emotion Recognition from Spatio-Temporal Representation of EEG Signals via 3D-CNN with Ensemble Learning Techniques/ Computer aided diagnosis system using deep convolutional neural networks for ADHD subtypes/ Multi-Kernel Temporal and Spatial Convolution for EEG-Based Emotion Classification
9- Make Figure 2 wider by redesigning them.
Reviewer 2 Report
The article "EEG Emotion Recognition by Fusion of Multi-scale Features" presents a new method for emotion recognition using electroencephalography (EEG) signals, namely the Multi-scale Deformable Convolutional Interacting Attention Network based on Residual Network (MDCNAResnet). This approach applies deformable convolution to extract abstract features and the Bottom-Up Feature Pyramid Network to capture multi-scale data features, with the integration of Efficient Channel Attention (ECANet) for feature extraction optimization. The model also incorporates a Bidirectional Gated Recurrent Unit (BiGRU) to capture the contextual semantic information of EEG signals. Testing on the DEAP dataset, the proposed method achieved high recognition accuracies of 98.63% for Valence and 98.89% for Arousal dimensions. Despite its successes, the authors recognized a need to simplify the network structure, reduce the number of parameters, and increase the efficiency of EEG signal recognition in future research. Upon reviewing this article, several critical weaknesses and limitations are evident as described below:
· The authors propose a Multi-scale Deformable Convolutional Interacting Attention Network based on Residual Network (MDCNAResnet) to improve EEG-based emotion recognition. However, they do not provide a clear rationale or explanation for the selection of this particular model or its components. It is uncertain whether these components were chosen due to proven effectiveness in the context of EEG data or because of a theoretical underpinning that justifies their use.
· The authors base their conclusions on the DEAP dataset exclusively, a relatively small dataset collected in a highly controlled environment. The lack of data diversity and the controlled conditions under which the data was collected severely limit the generalizability of the findings. The authors do not address this limitation nor do they validate their model on a more diverse dataset or in real-world settings, leading to potential overfitting and inflated performance metrics.
· The reported accuracies of 98.63% and 98.89% for Valence and Arousal dimensions, respectively, are surprisingly high. Given the complexity and individual variability inherent in EEG data, such high accuracies are unusual, raising questions about the robustness of the experiment. The authors do not discuss potential reasons for these unusually high accuracies nor do they validate their findings with additional metrics, such as precision, recall, or F1-score, which could provide a more comprehensive understanding of their model's performance.
· The article does not provide a comprehensive explanation of why differential entropy features were chosen for extraction, how they are particularly effective for this task, or how they compare to other possible features. This makes it difficult to evaluate the strength of their approach or replicate the study.
· The authors give sparse details about the implementation of their model and the hyperparameters used, reducing the reproducibility of the study. Additionally, the usage of only one GPU for the task may not be representative of real-world application scenarios where computational resources can be more constrained.
· There are no comparisons made with existing state-of-the-art methods. This makes it difficult to place their results in context and truly assess the novelty and effectiveness of their proposed model. The authors should consider more. Recognition of Human Inner Emotion Based on Two-Stage FCA-ReliefF Feature Optimization. Four-classes human emotion recognition via entropy characteristic and random forest. An Efficient Mixture Model Approach in Brain-Machine Interface Systems for Extracting the Psychological Status of Mentally Impaired Persons Using EEG Signals. Deep Convolutional Neural Network‐Based Visual Stimuli Classification Using Electroencephalography Signals of Healthy and Alzheimer’s Disease Subjects.
· The authors acknowledge the need to further simplify the network structure, reduce the number of parameters, and improve the efficiency of EEG signal recognition. This suggests that the current model might be too complex, possibly overfitting, and not optimized for real-world application.
Overall, while the authors' work represents an interesting exploration in the field of EEG-based emotion recognition, the aforementioned limitations diminish the scientific rigor and reliability of their study. Therefore, the presented work needs significant revisions to address these issues.
Round 2
Reviewer 1 Report
As a result of the authors' revisions, the paper's scientific level has increased. It appears that the paper is suitable for publication.
Author Response
I want to express my heartfelt gratitude for your approval of the revisions. Your feedback and endorsement mean a lot to me. Thank you.
Reviewer 2 Report
Many issues still remain in the revised version of the manuscript:
· Contributions are formulated, but what is the novelty?
· The explanation of ECA block is missing. Please provide a formal (mathematical) description.
· Figure 2 is not explained in the text.
· The accuracy, precision and recall measures are biased. It is suggested to provide F-score, at least.
· The quality of the figures must be increased. Figure 10 contains a text in Chinese language.
· Presentation of data in tables: units of measurement must be given (percents in Tables 2, 3, 4).
· The accuracy, precision and recall values are presented in percents, but equations (11-13) do not calculate values in percents.
· The analysis of the state-of-the-art is unsatisfactory.
Round 3
Reviewer 2 Report
The authors have addressed most of comments and improved. However, the comparison with state-of-the-art is still rather inadequate. The DEAP dataset is very popular, and many authors have used. Table 6 needs to enriched. Please consider DOI: 10.5755/j01.itc.51.1.29430. But there are also other EEG datasets apart from DEAP, see a review in DOI: 10.1049/pbhe019e_ch5. Why specifically the DEAP dataset was selected? Add an appropriate motivation.
Author Response
请参阅附件。
